# Random Entity Quantization for Parameter-Efficient Compositional Knowledge Graph Representation

**Jiaang Li[1], Quan Wang[2]\*, Yi Liu[3], Licheng Zhang[1], Zhendong Mao[1]**
[1]University of Science and Technology of China
[2]MOE Key Laboratory of Trustworthy Distributed Computing and Service,
Beijing University of Posts and Telecommunications
[3]State Key Laboratory of Communication Content Cognition
{jali,zlczlc}@mail.ustc.edu.cn, wangquan@bupt.edu.cn,
gavin1332@gmail.com, zdmao@ustc.edu.cn

## Abstract

Representation Learning on Knowledge Graphs (KGs) is essential for downstream tasks. The dominant approach, KG Embedding (KGE), represents entities with independent vectors and faces the scalability challenge. Recent studies propose an alternative way for parameter efficiency, which represents entities by composing entity-corresponding codewords matched from predefined small-scale codebooks. We refer to the process of obtaining corresponding codewords of each entity as entity quantization, for which previous works have designed complicated strategies. Surprisingly, this paper shows that simple random entity quantization can achieve similar results to current strategies. We analyze this phenomenon and reveal that entity codes, the quantization outcomes for expressing entities, have higher entropy at the code level and Jaccard distance at the codeword level under random entity quantization. Therefore, different entities become more easily distinguished, facilitating effective KG representation. The above results show that current quantization strategies are not critical for KG representation, and there is still room for improvement in entity distinguishability beyond current strategies. The code to reproduce our results is available here.

## 1 Introduction

Knowledge Graphs (KGs) comprise *(head entity, relation, tail entity)* triplets. They are crucial external knowledge sources for various natural language processing tasks (Hu et al., 2022; Sun et al., 2022). Learning representations on KGs is necessary for expressing complex semantics and supporting downstream tasks. The most dominant paradigm, KG Embedding (KGE), maps entities and relations to a vector space (Dettmers et al., 2018; Sun et al., 2019; Zhang et al., 2020). Despite the popularity, KGE models need to represent each

---
\*Corresponding author: Quan Wang

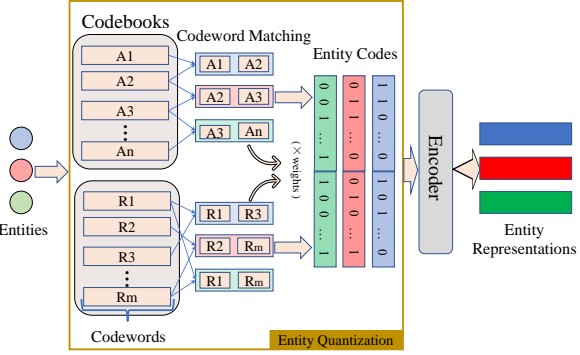

Figure 1: The process of parameter-efficient compositional KG representation. Entities are quantized to entity codes, which are encoded to represent entities. Each dimension of an entity code stands for a codeword, and indicates whether current entity matches this codeword (set to 1) with optional weights or not (set to 0).

entity with an independent vector, which leads to a linear increase in the number of parameters with the number of entities. Consequently, scalability becomes a challenge for these models, posing difficulties in their implementation and deployment (Peng et al., 2021; Ji et al., 2022), especially for large-scale KGs (Safavi and Koutra, 2020; Mahdisoltani et al., 2014).

A recently proposed parameter-efficient KG representation method uses compositional entity representations to reduce parameters (Galkin et al., 2022; Chen et al., 2023). Instead of learning separate representations like KGE, it represents entities by composing their matched codewords from predefined codebooks through an encoder, and requires fewer parameters since codewords are much fewer than entities. See Fig 1 for an illustration of this method. We refer to the process of obtaining corresponding codewords to each entity as entity quantization due to its similarity to vector quantization (van den Oord et al., 2017). Specifically, existing methods construct two codebooks, in which codewords are the entire relation set and a selec-

tive subset of entities, i.e., anchors, respectively. From these two codebooks, each entity matches two groups of codewords: connected relations and anchors nearby (Galkin et al., 2022) or with similar adjacent relations (Chen et al., 2023). Chen et al. (2023) also regards matching degrees as codeword weights for more expressiveness. Matched results are denoted as entity codes, including matched codewords and optional weights. A subsequent encoder uses entity codes to compose corresponding codewords and generate entity representations. This approach performs closely to KGE with fewer parameters, making KG training and deployment more efficient.

The key to entity quantization lies in two steps: (1) codebook construction and (2) codeword matching. Previous studies have dedicated their efforts to designing quantization strategies, which include selecting proper KG elements to construct codebooks and measuring the connectivity between codewords and entities to match them. We conduct experiments to randomize these strategies from shallow to deep. Surprisingly, we find that random entity quantization approaches can achieve similar or even better results.

We design several model variants for experiments. First, to investigate the effectiveness of matching codewords with connectivity, we randomize the codeword matching step by randomly selecting codewords as matched results of entities. Moreover, we set codeword weights randomly or equally for (Chen et al., 2023) to verify whether designed weights from matching are critical. Finally, to explore whether mapping codewords to actual elements in the KG is critical, we randomly construct codebooks with codewords that have no actual meaning. We adopt random codeword matching to the randomly constructed codebook, to provide a fully random entity quantization. Counterintuitively, empirical results show that the above operations achieve similar results compared to complicated quantization strategies and may even improve the model performance.

Moreover, we have verified that random entity quantization can better distinguish entities than current quantization strategies, which leads to more expressive KG representations (Zhang et al., 2022). Under the strategies designed by previous works, different entities could match the same codewords, making their code similar or identical. In contrast, random quantization leads to a lower possibility of matching same codewords and distributes entity codes more uniformly across a wide range. We prove this claim by analyzing the properties of entity codes. At the code level, we consider entity code as a whole and treat it as one sample of a random variable. The entropy of this variable can be derived from its distribution across all entity codes. We prove that random entity quantization has higher entropy and maximizes it with high probability, thus producing more diverse and unique entity codes. At the codeword level, each entity code indicates a set of matched codewords. We analyze the Jaccard distance between different sets and find that it is significantly increased by random entity quantization. As a result, different entities will have a more obvious dissimilarity when randomly quantized, making them easier to distinguish.

In summary, the contributions of our work are two-fold: (1) We demonstrate through comprehensive experiments that random entity quantization approaches perform similarly or even better than previously designed quantization strategies. (2) We analyze that this surprising performance is because random entity quantization has better entity distinguishability, by showing its produced entity codes have higher entropy and Jaccard distance. These results suggest that current complicated quantization strategies are not critical for the model performance, and there is potential for entity quantization approaches to increase entity distinguishability beyond current strategies.

## 2 Preliminaries

### 2.1 Knowledge Graph Representation

A knowledge graph $\mathcal{G} \subseteq \mathcal{E} \times \mathcal{R} \times \mathcal{E}$ is composed of entity-relation-entity triplets $(h, r, t)$, where $\mathcal{E}$ is a set of entities, and $\mathcal{R}$ is a set of relations. Each triplet indicates a relation $r \in \mathcal{R}$ between two entities $h, t \in \mathcal{E}$, where $h$ is the head entity, and $t$ is the tail entity. The goal of knowledge graph representation is to learn a vector representation $\mathbf{e}_i$ for each entity $e_i \in \mathcal{E}$, and $\mathbf{r}_j$ for relation $r_j \in \mathcal{R}$.

### 2.2 Compositional KG Representation

Compositional knowledge graph representation methods compose codewords from small-scale codebooks to represent entities. These methods obtain codewords for each entity by constructing codebooks and matching codewords. We refer to these two steps as entity quantization. The matched codewords are encoded to represent each entity.

This section presents two methods of this kind, i.e., NodePiece (Galkin et al., 2022) and EARL (Chen et al., 2023). We first introduce the definition of entity quantization and how these two methods represent entities with it. After that, we introduce how they are trained. Our subsequent experiments are based on these two methods.

### 2.2.1 Entity Quantization

We first formally define the entity quantization process. Existing entity quantization strategies construct a relation codebook $B_r = \{r_1, \cdots, r_m\}$ and an anchor codebook $B_a = \{a_1, \cdots, a_n\}$. The codewords $(r_1, \cdots, r_m)$ are all $m$ relations in $\mathcal{R}$ and $(a_1, \cdots, a_n)$ are $n$ anchors selected from all entities in $\mathcal{E}$ with certain strategies. After adding reverse edges to KG, each entity $e_i \in \mathcal{E}$ matches $s_i = min(d_i, d)$[1] unique relations from all its $d_i$ connected relations in $B_r$, and employs anchor-matching strategies to match $k$ anchors from $B_a$. Its matched codewords are denoted as a set $W_i = \{r_1^i, \cdots, r_{s_i}^i, a_1^i, \cdots, a_k^i\}$. Each entity $e_i$ will get its entity code $\mathbf{c}_i$ to represent $W_i$, which is a $(m+n)$-dimensional vector that is zero except for the $(s_i + k)$ dimensions representing matched codewords. Values in these dimensions are set to 1 or optional codeword weights if provided.

Then, we provide the detailed quantization process of both models.

**NodePiece** NodePiece uses metrics such as Personalized PageRank Page et al. (1999) to pick some entities as codewords in $B_a$. Each $e_i \in \mathcal{E}$ matches nearest anchors from $B_a$ as $\{a_1^i, \cdots, a_k^i\}$.

**EARL** EARL constructs $B_a$ with 10% sampled entities. Each entity $e_i \in \mathcal{E}$ matches anchors which have the most similar connected relations. Matched codewords are assigned to designed weights. Weights of $r \in \{r_1^i, \cdots, r_{s_i}^i\}$ are based on its connection count with $e_i$, and weights of each $a \in \{a_1^i, \cdots, a_k^i\}$ are based on the similarity between connected relation sets of $e_i$ and $a$.

After quantization, codewords in $W_i$ are composed by an encoder to output entity representation $\mathbf{e}_i$. The encoders of NodePiece and EARL are based on MLP and CompGCN (Vashishth et al., 2020), respectively.

### 2.2.2 Model Training

Here we introduce how to train both models. For each triplet $(h, r, t)$, representations of $h$ and $t$

---

[1] $d$ is a hyperparameter in Nodepiece, and is $+\infty$ in EARL.

are obtained from above. Each relation $r_j \in \mathcal{R}$ is represented independently. Both models use RotatE (Sun et al., 2019) to score triplets with $f(h, r, t) = -||\mathbf{h} \circ \mathbf{r} - \mathbf{t}||$, which maps entities and relations in complex space, i.e., $\mathbf{h}, \mathbf{r}, \mathbf{t} \in \mathbb{C}^d$.

NodePiece and EARL use different loss functions for different datasets, including binary cross-entropy (BCE) loss and negative sampling self-adversarial loss (NSSAL). For a positive triplet $(h, r, t)$, BCE loss can be written as:

$$\mathcal{L}_{BCE}(h, r, t) = -\log(\sigma(f(h, r, t)))$$
$$- \sum_{i=1}^{n} \log(1 - \sigma(f(h_i', r, t_i'))),$$

where $\sigma$ is the sigmoid function and $(h_i', r, t_i')$ is the $i$-th negative triplet for $(h, r, t)$.

NSSAL further considers that negative samples have varying difficulties:

$$\mathcal{L}_{NSSAL}(h, r, t) = -\log \sigma(\gamma - f(h, r, t))$$
$$- \sum_{i=1}^{n} p(h_i', r, t_i') \log \sigma(f(h_i', r, t_i') - \gamma),$$

where $\gamma$ is a fixed margin. $p(h_i', r, t_i')$ is the self-adversarial weight of $(h_i', r, t_i')$ and takes the following form:

$$p(h_j', r, t_j') = \frac{\exp \alpha(f(h_j', r, t_j'))}{\sum_i \exp \alpha f(h_i', r, t_i')},$$

where $\alpha$ is the temperature of sampling.

## 3 Experimental Setup

Though previous methods have designed complicated strategies for entity quantization, whether these strategies are critical for the model performance remains to be explored. Our experiments are to empirically test the effect of these quantization strategies. We therefore design a series of model variants using random entity quantization. We focus on showing the effectiveness of random entity quantization, rather than proposing new state-of-the-art models. Below we introduce the model variants we design, the datasets we test, and the training/evaluation protocols.

### 3.1 Model Variants

We design model variants based on existing models, NodePiece and EARL. We replace part of their designed quantization strategies with random approaches for each of these two models. The random

methods we use are generally divided into three types: (1) Randomly match entities to relations or anchors in codebooks. (2) Randomly or equally set codeword weights. (3) Randomly construct codebooks, where codewords do not refer to natural elements in the KG. We will discuss the details of these model variants in Section 4.

## 3.2 Datasets

We use three knowledge graph completion datasets in total. We employ FB15k-237 (Toutanova et al., 2015) and WN18RR (Dettmers et al., 2018) to demonstrate the effectiveness of random entity quantization and extend our conclusions to a larger scale KG, CoDEx-L(Safavi and Koutra, 2020). FB15k-237 is based on Freebase (Bollacker et al., 2008), a knowledge base containing vast factual information. WN18RR has derived from Wordnet (Miller, 1995), specifically designed to capture semantic relations between words. CoDEx-L is the largest version of recently proposed CoDEx datasets, which improve upon existing knowledge graph completion benchmarks in content scope and difficulty level. For consistency with compared methods, we exclude test triples that involve entities not present in the corresponding training set. Table 1 presents the statistics of these datasets. We also experiment with inductive relation prediction datasets(Teru et al., 2020) (details in Appendix C).

| Datasets | FB15k-237 | WN18RR | CoDEx-L |
|---|---|---|---|
| #Entity | 14,505 | 40,559 | 77,951 |
| #Relation | 237 | 11 | 69 |
| #Train | 272,115 | 86,835 | 551,193 |
| #Valid | 17,526 | 2,824 | 30,622 |
| #Test | 20,438 | 2,924 | 30,622 |

Table 1: Statistics of the benchmark datasets, including the number of entities, relations, training triples, validation triples, and test triples.

## 3.3 Training and Evaluation Protocols

**Training.** Our experiments are based on the official implementation of NodePiece and EARL. We use the same loss functions and follow their hyperparameter setting for corresponding model variants. More details are provided in Appendix A.

**Evaluation.** We generate candidate triplets by substituting either $h$ or $t$ with candidate entities for each triplet $(h, r, t)$ in the test sets. The triples are then sorted in descending order based on their scores. We apply the filtered setting (Bordes et al., 2013) to exclude other valid candidate triplets from ranking. To assess the performance of the models, we report the mean reciprocal rank (MRR) and Hits@10. Higher MRR/H@10 indicates better performance. Additionally, we evaluate the efficiency of the models using $Effi = MRR/\#P$, where $\#P$ represents the number of parameters. The results of NodePiece and EARL are from their original papers.

## 4 Random Entity Quantization

This section details the random variants we design and their experimental results. We design model variants by randomizing different steps of existing entity quantization strategies, including codeword matching and codebook construction. We find that these variants achieve similar performance to existing quantization strategies. These results suggest that the current entity quantization strategies are not critical for model performance.

### 4.1 Random Codeword Matching

We first randomize the codeword matching step, which includes strategies for (1) matching each entity to the corresponding codewords and (2) assigning weights to the matched codewords.

### 4.1.1 Matching Strategy

We randomize the codeword matching strategies to investigate whether current connectivity-based strategies are critical to the model performance. We design model variants by randomizing current methods' relation-matching or anchor-matching strategies and keep other settings unchanged. Specifically, with the relation codebook $B_r$ and the anchor codebook $B_a$, we have the following model variants for both models.

- +RSR: Each entity $e_i \in \mathcal{E}$ **R**andomly **S**elects $s_i$ **R**elations (RSR) from $B_r$ and matches $k$ anchors from $B_a$ with the anchor-matching strategy designed by the current model, as matched codewords $W_i$.

- +RSA: $e_i$ **R**andomly **S**elects $k$ **A**nchors (RSA) from $B_a$, and matches $s_i$ relations from $B_r$ with current relation-matching strategy, as $W_i$.

- +RSR+RSA: $e_i$ randomly selects $s_i$ relations from $B_r$, and randomly selects $k$ anchors from $B_a$, as $W_i$.

| | FB15k-237 | | WN18RR | |
|---|---|---|---|---|
| | MRR | Hits@10 | MRR | Hits@10 |
| EARL | 0.310 | 0.501 | 0.440 | 0.527 |
| +RSR | 0.306 | 0.500 | 0.439 | 0.530 |
| +RSA | 0.311 | 0.506 | 0.438 | 0.529 |
| +RSR+RSA | 0.308 | 0.502 | 0.442 | 0.536 |
| NodePiece | 0.256 | 0.420 | 0.403 | 0.515 |
| +RSR | 0.254 | 0.417 | 0.403 | 0.516 |
| +RSA | 0.258 | 0.423 | 0.419 | 0.518 |
| +RSR+RSA | 0.263 | 0.425 | 0.425 | 0.522 |

Table 2: Results for parameter-efficient compositional KG representation methods with randomly selected relations (RSR) or randomly selected anchors (RSA).

For all variants, we still assign codewords in $W_i = \{r_1^i, \cdots, r_{s_i}^i, a_1^i, \cdots, a_k^i\}$ with current connectivity-based weights, if required.

Table 2 shows the performance of original models and their respective variants. Surprisingly, randomizing codeword-matching and relation-matching does not affect the overall performance of existing methods on both datasets, whether used together or separately. The results suggest that current complicated codeword matching strategies are not critical to the model performance.

We further study the impact of randomizing the codeword matching strategy with only one codebook. We remove $B_r$ or $B_a$ respectively, and adopt different codeword matching approaches for the remaining codebook, forming following variants:

- w/o anc: Remove the anchor codebook $B_a$. Match $s_i$ relations from $B_r$ with the current relation-matching strategy as $W_i = \{r_1^i, \cdots, r_{s_i}^i\}$.

- w/o anc+RSR: Remove $B_a$. Randomly select $s_i$ relations from $B_r$ as $W_i = \{r_1^i, \cdots, r_{s_i}^i\}$.

- w/o rel: Remove the relation codebook $B_r$. Match $k$ anchors from $B_a$ with the current anchor-matching strategy as $W_i = \{a_1^i, \cdots, a_k^i\}$.

- w/o rel+RSA: Remove $B_r$. Randomly select $k$ anchors from $B_a$ as $W_i = \{a_1^i, \cdots, a_k^i\}$.

Table 3 shows that when removing one codebook, random matching codewords from the remaining codebook performs better than using current designed matching strategies. It even performs similarly to the original methods in most

| | FB15k-237 | | WN18RR | |
|---|---|---|---|---|
| | MRR | Hits@10 | MRR | Hits@10 |
| EARL | 0.310 | 0.501 | 0.440 | 0.527 |
| w/o anc | 0.301 | 0.488 | 0.409 | 0.498 |
| w/o anc+RSR | 0.312 | 0.501 | 0.417 | 0.516 |
| w/o rel | 0.309 | 0.501 | 0.432 | 0.520 |
| w/o rel+RSA | 0.311 | 0.500 | 0.443 | 0.539 |
| NodePiece | 0.256 | 0.420 | 0.403 | 0.515 |
| w/o anc | 0.204 | 0.355 | 0.011 | 0.019 |
| w/o anc+RSR | 0.244 | 0.409 | 0.009 | 0.014 |
| w/o rel | 0.258 | 0.425 | 0.266 | 0.465 |
| w/o rel+RSA | 0.256 | 0.428 | 0.411 | 0.517 |

Table 3: Random codeword matching with only one codebook. 'w/o anc' denotes not using the anchors, and 'w/o rel' denotes not using the relation codebook. Ablation results are taken from the original paper.

| | FB15k-237 | | WN18RR | |
|---|---|---|---|---|
| | MRR | Hits@10 | MRR | Hits@10 |
| EARL | 0.310 | 0.501 | 0.440 | 0.527 |
| +RW | 0.308 | 0.498 | 0.442 | 0.531 |
| +EW | 0.308 | 0.500 | 0.437 | 0.528 |

Table 4: Results for parameter-efficient compositional KG representation methods with random codeword weights (RW) or equal codeword weights (UW)

cases. NodePiece performs poorly on WN18RR without $B_a$, because the number of relations in this dataset is small, and only using $B_r$ sharply decreases model parameters. The above results further validate the effectiveness of random codeword matching and demonstrate its robustness with only one codebook.

### 4.1.2 Codeword Weights from Matching

During the matching process, EARL will further assign weights to the matched codewords based on the connectivity between entities and codewords. We conduct experiments to explore whether such weights are critical. Specifically, we design following model variants for EARL using random or equal codeword weights, with the codebooks and codeword matching strategies unchanged:

- +RW: Assign **R**andom **W**eights (RW) to matched codewords.

- +EW: Assign **E**qual **W**eights (EW) to matched codewords, i.e., set all codeword weights to 1.

| | FB15k-237 | | | | WN18RR | | | | CoDEx-L | | | |
|---|---|---|---|---|---|---|---|---|---|---|---|---|
| | #P(M) | MRR | Hits@10 | Effi | #P(M) | MRR | Hits@10 | Effi | #P(M) | MRR | Hits@10 | Effi |
| EARL | 1.8 | 0.310 | 0.501 | 0.172 | 3.8 | 0.440 | 0.527 | 0.116 | 2.1 | 0.238 | 0.390 | 0.113 |
| EARL+RQ | 1.4 | **0.311** | **0.505** | **0.222** | 3.3 | **0.444** | **0.535** | **0.135** | 1.9 | **0.239** | **0.394** | **0.126** |
| NodePiece | 3.2 | 0.256 | 0.420 | 0.080 | 4.4 | 0.403 | 0.515 | 0.092 | 3.6 | 0.190 | 0.313 | **0.053** |
| NodePiece+RQ | 3.2 | **0.261** | **0.423** | **0.082** | 4.4 | **0.429** | **0.517** | **0.098** | 3.6 | **0.192** | **0.326** | **0.053** |

Table 5: Results of applying fully random entity quantization (RQ) to parameter-efficient compositional KG representation methods on datasets with varying sizes. Better results between each model and its variant are bolded.

Table 4 shows that either using random weights or equal weights does not significantly impact the performance. Thus, we can conclude that the connectivity-based weights are not critical for model performance. For subsequent experiments, we use equal codeword weights for simplicity.

## 4.2 Random Codebook Construction

We construct codebooks randomly to investigate whether it is critical to construct codebooks with specific relations or entities as codewords, like current strategies. We design a model variant for each model, which uses a randomly constructed codebook $B$ instead of $B_r$ and $B_a$. Codewords in $B$ do not have any real meanings. Moreover, we adopt random codeword matching and equal codeword weights to such variants. We call this type of variant the fully random entity quantization (RQ) since it randomizes all parts of the entity quantization.

- +RQ: Fully **R**andom entity **Q**uantization (RQ). Randomly construct a codebook whose size equals the sum of $B_r$ and $B_a$: $B = \{z_1, \ldots, z_{m+n}\}$. Each entity $e_i \in \mathcal{E}$ randomly selects $(s + k)$ codewords as its matched codewords $W_i = \{z_1^i, \ldots, z_{s+k}^i\}$ with equal weights, where $s = \frac{1}{|\mathcal{E}|} \sum_{i=1}^{|\mathcal{E}|} s_i$ and $|\mathcal{E}|$ is the number of entities.

Table 5 shows variants of both models lead to similar or even better performance across all datasets with varying sizes. We will analyze this surprising observation in Section 5. Through $\#P(M)$ and $Effi$, we further show that random entity quantization requires equal or even fewer parameters with higher efficiency. It does not require composing each codebook's codewords separately, saving parameters for EARL. The above results show that constructing codebooks by KG elements is not critical to the model performance. From all variants and results above, we can conclude that random entity quantization is as effective as current complicated quantization strategies.

## 5 Why Random Quantization Works

This section analyzes the reasons for the surprising performance of random entity quantization. The entity codes directly affect entity representation and model performance. We analyze the entity codes produced by different entity quantization approaches to compare their ability to distinguish different entities. We find random entity quantization obtains entity codes with greater entropy at the code level and Jaccard distance at the codeword level. Thus, it can better distinguish different entities and represent KGs effectively.

### 5.1 Code Level Distinguishability

We analyze the ability to distinguish entities at the code level of different entity quantization approaches. Specifically, we treat the entity code $\mathbf{c}_i$ of each entity $e_i \in \mathcal{E}$ as a sampling of a random variable $X$. The entity codes of all entities represent $|\mathcal{E}|$ samplings of $X$. From these samplings, we get $v$ different entity codes and their frequencies. We denote these codes as $\{x_1, \ldots, x_v\}$ and their frequencies as $\{f_1, \ldots, f_v\}$. We denote $l = m + n$ as the total number of codewords, where $m$ and $n$ are numbers of codewords in $B_r$ and $B_a$. The number of all possible entity codes is $2^l$. We estimate the probability distribution of $X$ on all codes based on the relative frequency of different entity codes in the sampling results, and then derive its entropy:

$$H(X) = -\sum_{i=1,\ldots,2^l} P(x_i) \cdot \log_2 P(x_i), \quad (1)$$

where $P(x_i)$ is the relative frequency of $x_i$:

$$P(x_i) = \begin{cases} \frac{f_i}{|\mathcal{E}|} & \text{if} \quad i = 1, \ldots, v, \\ 0 & \text{if} \quad i = v+1, \ldots, 2^l. \end{cases}$$

|          | NodePiece | EARL  | Random |
|----------|-----------|-------|--------|
| FB15k-237 | 15.26    | 14.50 | 15.27  |
| WN18RR    | 15.94    | 8.20  | 16.75  |

Table 6: The entropy (bits) of entity code produced by random entity quantization and well-designed quantization strategies proposed by NodePiece and EARL.

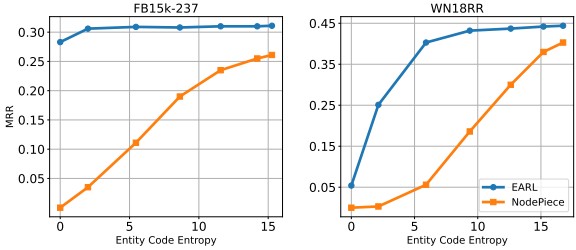

Figure 2: Performance of random entity quantization with different entropy

We use the entropy $H(X)$ in eq. 1 to measure the diversity of entity codes. A higher entropy means more diverse entity codes, thus indicating the better entity-distinguish ability of the quantization approach. In this sense, this entropy can reflect entity distinguishability at the code level.

We use equation 1 to derive the entity code entropy produced by random and previously designed quantization strategies. The results are listed in Table 6. The entropy of random quantization is averaged in 100 runs with different seeds, while the entropy of previous strategies is deterministic and does not change in different runs. Empirically, we demonstrate that random quantization achieves higher entropy, producing more diverse entity codes and enabling easier distinguishability between different entities.

We confirm that higher entity code entropy brings better model performance through extensive experiments. Specifically, after random entity quantization, we randomly select a subset of entity codes and set them to be identical, to obtain entity codes with different entropy values. Figure 2 shows the comparison experiments across these entity codes with EARL and NodePiece. As the entropy rises, the model performance gradually increases and eventually saturates. EARL performs better than NodePiece across varying entropy, as its GNN encoder involves the entity neighborhood and can better distinguish entities. From above, more diverse entity codes with higher entropy benefit the model performance, which aligns with our claims.

In addition, the entity code entropy is maximized when all entity codes are unique. The probability of random entity quantization to produce $|\mathcal{E}|$ unique entity codes is close to 1, as shown in Theorem 1. The detailed proof is in Appendix B. This theorem shows that random entity quantization expresses entities uniquely with high probability, thus distinguishing entities well.

**Theorem 1.** *The probability of random entity quantization to produce $|\mathcal{E}|$ unique entity codes is $P = \prod_{i=0}^{|\mathcal{E}|-1} \frac{2^l - i}{2^l}$, which approaches 1 when $2^l \gg |\mathcal{E}|$.*

From above, we demonstrate random entity quantization produces diverse entity codes and clearly distinguishes entities at the code level.

### 5.2 Codeword Level Distinguishability

We further analyze the ability to distinguish entities at the codeword level of different entity quantization approaches. Specifically, for entities $e_i, e_j \in \mathcal{E}$ with entity codes $\mathbf{c}_i$ and $\mathbf{c}_i$, their corresponding sets of matched codewords are $W_i = \{r_1^i, \cdots, r_{s_i}^i, a_1^i, \cdots, a_k^i\}$ and $W_j = \{r_1^j, \cdots, r_{s_j}^j, a_1^j, \cdots, a_k^j\}$. The Jaccard distance between $\mathbf{c}_i$ and $\mathbf{c}_i$ is:

$$d_J(\mathbf{c}_i, \mathbf{c}_j) = \frac{|W_i \cup W_j| - |W_i \cap W_j|}{|W_i \cup W_j|},$$

where $|\cdot|$ denotes the number of elements in a set.

We use the Jaccard distance $d_J(\mathbf{c}_i, \mathbf{c}_j)$ to measure the distinctiveness between entity codes $\mathbf{c}_i$ and $\mathbf{c}_j$. A larger distance means their indicated codewords are more distinct and makes entities $e_i$ and $e_j$ more easily to be distinguished. In this sense, this distance can reflect the entity distinguishability at the codeword level.

To capture the overall distinguishability among all entity codes, we propose a $k$-nearest neighbor evaluation metric based on the Jaccard distance. This evaluation assesses the average distance between each entity code and its $k$ nearest codes, denoted as $\mathcal{J}_k$. A higher $\mathcal{J}_k$ means the entity codes are more distinct. We use different values of $k$ to observe the distance among entity codes in different neighborhood ranges. The metric $\mathcal{J}_k$ is derived as:

$$\mathcal{J}_k = \frac{1}{|\mathcal{E}| \times k} \sum_{e_i \in \mathcal{E}} \sum_{e_j \in kNN(e_i)} d_J(\mathbf{c}_i, \mathbf{c}_j),$$

where $|\mathcal{E}|$ is the number of entities. $kNN(e_i)$ is a set of $k$ entities whose codes are nearest to the

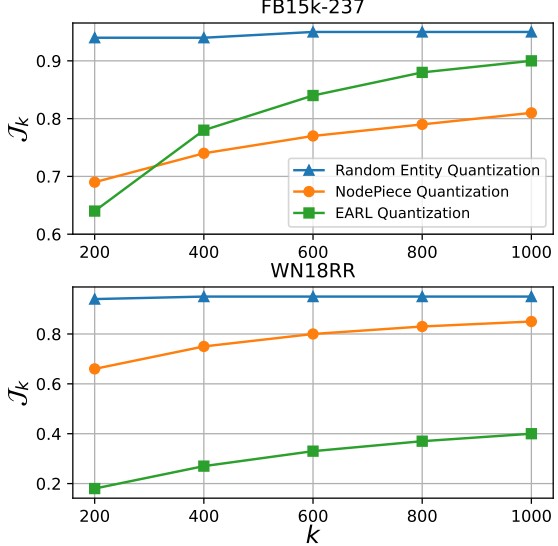

Figure 3: The average Jaccard distance between each entity code and its $k$ nearest codes.

code of $e_i$ under Jaccard distance:

$$kNN(e_i) = \underset{\substack{\{e_{l_1},...,e_{l_k}\} \subset \mathcal{E} \\ e_{l_1},...,e_{l_k} \neq e_i}}{\arg\min} \sum_{j \in \{l_1,...,l_k\}} d_J(\mathbf{c}_i, \mathbf{c}_j).$$

Figure 3 shows the average Jaccard distance $\mathcal{J}_k$ between entity codes w.r.t. different numbers $k$ of nearest codes. We can see that random entity quantization achieves higher $\mathcal{J}_k$ than current quantization strategies across the varying $k$. Thus, entity codes produced by random entity quantization are more distinct within different neighborhood ranges. Based on the above observation, random entity quantization makes different entities easier to distinguish at the codeword level.

## 5.3 Discussion

We can derive from the above that, entity distinguishability is the reason why current quantization strategies based on attribute similarity don't work better than the random approach. From Table 6 and Figure 3, it's proved that random entity quantization distinguishes entities better in both code and codeword levels. Furthermore, in Figure 2, we show that entity quantization strategies with higher entity distinguishability tend to perform better. Therefore, it's reasonable that random quantization works comparable to current strategies or even better.

## 6 Related Work

**Knowledge Graph Embedding** KG embedding aims at learning independent representations for entities and relations of a given KG, which benefits downstream tasks such as question answering (Hu et al., 2022), reading comprehension (Sun et al., 2022), and pre-trained language representation Wang et al., 2021c. Recent years have witnessed a growing number of KG embedding techniques being devised, including distance-based models (Bordes et al., 2013; Sun et al., 2019; Zhang et al., 2020), semantic matching models (Trouillon et al., 2016; Balažević et al., 2019), neural encoding models (Dettmers et al., 2018; Schlichtkrull et al., 2018; Wang et al., 2019), and text augmented models (Yao et al., 2019; Wang et al.). We refer readers to (Wang et al., 2017; Ji et al., 2022) for a comprehensive overview of the literature.

**Parameter-Efficient KG Representation** KG embedding methods face the scalability challenge. The number of parameters scales up linearly to the entity number. Several studies compress learned parameters from KG embedding models, trying to solve this issue. Incorporating knowledge distillation techniques, MulDE (Wang et al., 2021b) transfers knowledge from multiple teacher models to a student model. Expanding on this, DualDE (Zhu et al., 2022) considers the dual influence between the teacher and student and adapts the teacher to better align with the student, thus enhancing the performance of the distilled model. To directly compress existing models, Sachan (2020) discretizes the learned representation vectors for less parameter storage space, then maps the discrete vectors back to the continuous space for testing. LightKG (Wang et al., 2021a) introduces dynamic negative sampling and discretizes learned representations through vector quantization.

However, the above methods firstly need to train KG embedding models with full parameter size. Recently proposed compositional parameter-efficient KG representation models (Galkin et al., 2022; Chen et al., 2023), which are illustrated in this paper, enable a more efficient training process.

**Random Features in Graph Representation** In homogeneous graph learning, researchers prove that message-passing neural networks are more powerful when having randomly initialized node features (Sato et al., 2021; Abboud et al., 2021). In KGs, Zhang et al. (2022) finds that random perturbation of relations does not hurt the performance of graph convolutional networks (GCN) in KG completion. Degraeve et al. (2022) further implies

that an R-GCN with random parameters may be comparable to an original one. To the best of our knowledge, there is no existing study on representing entities with randomness in parameter-efficient compositional KG representation.

# 7 Conclusion

In conclusion, this paper demonstrates the effectiveness of random entity quantization in parameter-efficient compositional knowledge graph representation. We explain this surprising result by illustrating that random quantization could distinguish different entities better than current entity quantization strategies. Thus, we suggest that existing complicated entity quantization strategies are not critical for model performance, and there is still room for entity quantization approaches to improve entity distinguishability beyond these strategies.

## Limitations

This paper only studies entity quantization with encoders proposed in early works, while designing more expressive encoders is also important and can improve the performance of parameter-efficient compositional knowledge graph representation. We leave this part as future work.

## Acknowledgments

We would like to thank Ruichen Zheng for the discussion on measuring distinguishability, Zhengbo Wang, Haotian Zhang, Chengchao Xu for the discussion on entropy computation, and Jiahao Li for help to improve the technical writing of this paper.

This work is supported by the National Key Research and Development Program of China under Grant 2021YFF0901600, the National Science Fund for Excellent Young Scholars under Grant 62222212, and the National Natural Science Foundation of China under Grant 62376033.

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

## A  Training Details

We use the exact same hyperparameter settings as in the original papers of NodePiece and EARL. The results of model variants with random approaches are averaged in three runs with different seeds. We do not further tune hyperparameters for our proposed model variants. Thus, their performance may be underestimated. But so long as the model variants perform similarly to the original models, we can still make our conclusions. For NodePiece, we obtain matched anchors with its 'shortest-path' mode to be consistent with the original paper. For EARL, we use its released anchor set. We run our experiments with a single RTX 3090 GPU with 24GB RAM.

We use uniformly distributed random numbers in our model variants. It means that in model variants that randomly match entities to codewords, each codeword has an equal probability of being matched.

## B  Proof of Theorem 1

*Proof.* Since random entity quantization matches entities to codewords with equal probabilities, it produces all entity codes uniformly. Thus, the probability of random entity quantization to get $|\mathcal{E}|$ different entity codes is:

$$Prob = \frac{P(2^l, |\mathcal{E}|)}{(2^l)^{|\mathcal{E}|}} = \frac{\frac{2^l!}{(2^l-|\mathcal{E}|)!}}{(2^l)^{|\mathcal{E}|}} = \prod_{i=0}^{|\mathcal{E}|-1} \frac{2^l - i}{2^l}$$

where $P(n, r)$ is the number of permutations of selecting $r$ elements from a set of $n$ elements. $l$ is the total number of codewords, and the total number of entity codes that can be produced is $2^l \gg |\mathcal{E}|$. □

## C  Inductive Relation Prediction Results

Besides FB15K-237 and WN18RR datasets used in the main text, we further test the effectiveness of random entity quantization on inductive relation prediction, where NodePiece has shown superiority. This task requires learning from one KG, and generalizes to another KG with no shared entities for inference. We follow previous works (Zhu et al., 2021; Galkin et al., 2022; Li et al., 2023) and use the datasets proposed by Teru et al. (2020), including four versions of subsets generated from FB15k-237. We test the performance of NodePiece with the fully random entity quantization (RQ) as

| | FB15K-237 | | | |
| --- | --- | --- | --- | --- |
| | v1 | v2 | v3 | v4 |
| NodePiece | 0.873 | 0.939 | 0.944 | 0.949 |
| NodePiece+RQ | 0.867 | 0.942 | 0.945 | 0.944 |

Table 7: Inductive relation prediction results (Hits@10) of NodePiece and its variant with fully random entity quantization (RQ).

described in Section 4.2, on these subsets. We set the learning rate to 1e-3 and train the variant with 300 epochs on v1/v4, and 120 epochs on v2/v3. The other hyperparameters remain the same as the original method. We use the exact same evaluation protocol as in previous works (Teru et al., 2020, Zhu et al., 2021; Galkin et al., 2022; Li et al., 2023). The results are shown in Table 7.

We can see that NodePiece's variants with random entity quantization perform as well as the original model in inductive link prediction. The results support and strengthen our claim that random entity quantization is effective, both in the transductive and inductive settings.