# OpenReview forum: "Random Entity Quantization for Parameter-Efficient Compositional  Knowledge Graph Representation"
_EMNLP/2023/Conference — EMNLP 2023 Main_

### Official Review · Reviewer_Y2GK · 2023-08-03

**Soundness:** 4

**Excitement:**

4: Strong: This paper deepens the understanding of some phenomenon or lowers the barriers to an existing research direction.

**Paper Topic And Main Contributions:**

This work revisits the current compositional knowledge graph representation methods, and experimentally validate that simple random entity quantization can achieve state-of-the-art performance. It further justifies such phenomenon by revealing that the random quantization has high entropy at the code level and Jaccard distance at the codeword level, which makes entity more distinguishable.

**Questions For The Authors:**

Q1 In Section 3.3, is the evaluation setting the same as the state-of-the-art works?
Q2 Is there any further research direction?

**Reasons To Accept:**

S1. The paper is well motivated and clearly organized.
S2. The claim is well supported by experimental studies and the theoretical analyses.

**Reasons To Reject:**

Only some minor issues. See below.

**Reproducibility:**

4: Could mostly reproduce the results, but there may be some variation because of sample variance or minor variations in their interpretation of the protocol or method.

**Reviewer Confidence:**

3: Pretty sure, but there's a chance I missed something. Although I have a good feel for this area in general, I did not carefully check the paper's details, e.g., the math, experimental design, or novelty.

---

> ### Author Rebuttal · Authors · 2023-08-28
>
> Thanks for the reviewer’s positive evaluation, including well motivated, clearly organized, and well supported. There are also valuable questions we try to answer as follows.
>
>
>
> Q1: Is the evaluation setting the same as the state-of-the-art works?
>
> A1: Yes, we use evaluation settings that are widely used in many state-of-the-art works, such as [1]\[2\][3].
>
>
>
> Q2: Is there any further research direction?
>
> A2: Based on our work, we think further research could study how to quantize entities to make them more distinguishable in theory. It would also be interesting to design more powerful encoders to boost the model’s performance.
>
>
>
> [1] RotatE: Knowledge Graph Embedding by Relational Rotation in Complex Space.
>
> [2] Learning Hierarchy-Aware Knowledge Graph Embeddings for Link Prediction.
>
> [3] Entity-agnostic representation learning for parameter-efficient knowledge graph embedding.

---

### Official Review · Reviewer_B12b · 2023-08-05

**Typos Grammar Style And Presentation Improvements:** 1. Usage of the term "quantization" i…
**Soundness:** 4

**Excitement:**

3: Ambivalent: It has merits (e.g., it reports state-of-the-art results, the idea is nice), but there are key weaknesses (e.g., it describes incremental work), and it can significantly benefit from another round of revision. However, I won't object to accepting it if my co-reviewers champion it.

**Paper Topic And Main Contributions:**

This paper conducts an empirical analysis of random entity quantization for compositional knowledge graph embedding (KGE) methods. Two baseline methods, NodePiece and EACL are selected. The results show that random entity quantization can achieve similar performance as current entity quantization strategies due to high entropy in the code level, which makes the entities distinguishable.

**Reasons To Accept:**

1. The analysis is interesting and important for future research in compositional KGE.
2. The conclusion suggests current entity quantization strategies can be improved, which encourages more research in this direction.

**Reasons To Reject:**

1. While it's explained in the paper why random entity quantization works, it's not well-explained why current quantization strategies that are based on PageRank or attribute similarity don't work better than random entity quantization. A general direction for a better entity quantization strategy is also desired in such an analysis.
2. The method is not really parameter-efficient. The parameter saving in Table. 5 is not significant.
3. The advantages of compositional KGE methods are in inductive link prediction, but the experiments are all transductive link prediction (as I can tell from the paper). Thus, the results are not convincing that random entity quantization is really better or as good.

======= After revision =======

Thanks to the authors for the comprehensive response. It addressed the majority of my questions and concerns. The additional experiment on inductive link prediction is also encouraging. Thus, I decided to raise the Soundness to 4 (strong) and Excitement to 3 (Ambivalent).

**Reproducibility:**

4: Could mostly reproduce the results, but there may be some variation because of sample variance or minor variations in their interpretation of the protocol or method.

**Reviewer Confidence:**

4: Quite sure. I tried to check the important points carefully. It's unlikely, though conceivable, that I missed something that should affect my ratings.

---

> ### Author Rebuttal · Authors · 2023-08-28
>
> Thanks for a valuable and constructive review. We will try to answer all the questions below, and hopefully address the reviewer's concerns.
>
>
>
> Q1: While it's explained in the paper why random entity quantization works, it's not well-explained why current quantization strategies that are based on PageRank or attribute similarity don't work better than random entity quantization. A general direction for a better entity quantization strategy is also desired in such an analysis.
>
> A1: The reason why current quantization strategies don’t work better than random entity quantization is that random entity quantization can distinguish different entities better, which benefits the model performance. To prove that, we kindly refer the reviewer to Table 6 and Figure 3, where we show that random entity quantization has higher entity distinguishability. For example, it achieves 16.75 bits in code-level entropy on the WN18RR dataset, while the strategy of EARL only reaches 8.20 bits. Furthermore, in Figure 2, we show that entity quantization strategies with higher entity distinguishability tend to perform better and achieve higher MRR. Therefore, it's reasonable that random entity quantization works comparable to current strategies or even better.
>
> Based on our work, we think a general direction for a better entity quantization strategy could be designing strategies to further increase the distinguishability of different entities, which would make models perform better.
>
> We will add more discussions to clarify the above points during the revision.
>
>
>
> Q2: The method is not really parameter-efficient. The parameter saving in Table. 5 is not significant.
>
> A2: Our claim is not to save more parameters than existing compositional KGE methods (EARL/NodePiece), which our method builds on. They are already parameter-efficient compared to traditional KGE methods like RotatE, as supported by the comparison in parameter sizes in the table below.
>
>  Our purpose in listing the parameter sizes in Table 5 is to show that, although it's not our initial intention, our proposed random entity quantization is even more efficient than existing compositional methods. For example, EARL has 0.527 Hits@10 with 3.8M parameters on WN18RR, while with fully random entity quantization, it can achieve 0.535 Hits@10 with only 3.3M parameters.
>
> | Method    | FB15k-237 | WN18RR | CoDEx-L |
> | --------- | --------- | ------ | ------- |
> | RotatE    | 29.3 M    | 40.6 M | 78 M    |
> | EARL      | 1.8 M     | 3.8 M  | 2.1 M   |
> | NodePiece | 3.2 M     | 4.4 M  | 3.6 M   |
>
>
>
>  Q3: The advantages of compositional KGE methods are in inductive link prediction, but the experiments are all transductive link prediction (as I can tell from the paper). Thus, the results are not convincing that random entity quantization is really better or as good.
>
> A3: We have conducted the inductive link prediction experiments to make this a more comprehensive work. We use inductive link prediction datasets proposed in [1], including four versions of subsets based on FB15k-237 (FB237). We test NodePiece with the completely random quantization (RQ) proposed in our paper, and show the preliminary results (Hits@10) in the table below. We will complete these results during paper revision due to the limited time.
>
> The results show that the compositional KGE method with random entity quantization performs as well as the original model in inductive link prediction, which supports and strengthens our claim that random entity quantization works as well as current strategies.
>
> | Method       | FB237-v1 | FB237-v2 | FB237-v3 | FB237-v4 |
> | ------------ | -------- | -------- | -------- | -------- |
> | NodePiece    | 0.873    | 0.939    | 0.944    | 0.949    |
> | NodePiece+RQ | 0.867    | 0.942    | 0.945    | 0.944    |
>
>
>
> Q4: Usage of the term "quantization" is misleading.
>
> A4: We model compositional KGE methods as "entity quantization" because we think this term could appropriately describe the process of such methods and their similarity to vector quantization. We kindly refer the reviewer to line 48 for details about entity quantization. In short, like the way to represent vectors in vector quantization, entities, without independent representations, are represented by codewords matched from the codebooks. Another similarity is that both techniques save parameters due to this kind of representation. Therefore, we choose this term to highlight the characteristics of compositional KGE methods. We will add more discussions during the paper revision to clarify the above points.
>
>
>
> Q5: Best performance can be marked in boldface for clarity in Table. 2, 3, and 4.
>
> A5: Thanks for your kind advice. The purpose of these tables is to show that different variants of random entity quantization perform comparable to the original method, rather than highlight the highest value. So we didn't bold the best performance.
>
>
>
> [1] Inductive Relation Prediction by Subgraph Reasoning.

---

### Official Review · Reviewer_ummp · 2023-08-06

**Soundness:** 3

**Excitement:**

4: Strong: This paper deepens the understanding of some phenomenon or lowers the barriers to an existing research direction.

**Missing References:**

None.

**Paper Topic And Main Contributions:**

This study investigates entity quantization in parameter-efficient knowledge graph (KG) embedding models, which is the process of getting corresponding codewords for each entity. It presents a random entity quantization strategy and creates several model variants based on three strategies for obtaining the matching codewords for each entity: randomly choosing codewords, randomly or equally setting codeword weights, and randomly generating codebooks. The authors conducted extensive experiments to compare the proposed quantization strategies to previously designed quantization methods. The experimental results reveal an intriguing finding: random quantization can yield results comparable to existing complex quantization methods. Analyses are also offered to back up the findings. This study, I think, can help advance the development of parameter-efficient KG embedding models.

**Questions For The Authors:**

None.

**Reasons To Accept:**

- This paper points out the shortcomings of existing parameter-efficient KG embedding models and will inspire future work in this field.

- The idea of modeling existing entity composition strategies as random entity quantization is novel.

- The authors conducted expensive experiments to demonstrate the effectiveness of the proposed method and show an interesting finding that random quantization can yield results comparable to existing quantization methods.

**Reasons To Reject:**

- Intuitively, parameter-efficient KG embedding models can perform inductive inference. NodePiece conducts experiments on inductive link prediction. It is better to provide results about inductive link prediction to further demonstrate the effectiveness of the proposed random entity quantization.

- In line 178, the authors mentioned that one entity matches all its connected relations. However, NodePiece randomly samples a set of $m$ unique outgoing relations for an entity, where $m$ is a hyper parameter. It seems that the statement in this paper is in conflict with that in NodePiece.

**Reproducibility:**

4: Could mostly reproduce the results, but there may be some variation because of sample variance or minor variations in their interpretation of the protocol or method.

**Reviewer Confidence:**

4: Quite sure. I tried to check the important points carefully. It's unlikely, though conceivable, that I missed something that should affect my ratings.

**Typos Grammar Style And Presentation Improvements:**

- L262: CoDEx-L(Safavi and Koutra, 2020) -> CoDEx-L (Safavi and Koutra, 2020)

---

> ### Author Rebuttal · Authors · 2023-08-28
>
> We thank the reviewer for a positive evaluation, including novel, inspiring, interesting, and extensive experiments, etc. There are also valuable questions and advice we try to answer as follows.
>
> Q1: Intuitively, parameter-efficient KG embedding models can perform inductive inference. NodePiece conducts experiments on inductive link prediction. It is better to provide results about inductive link prediction to further demonstrate the effectiveness of the proposed random entity quantization.
>
> A1: We have conducted the inductive link prediction experiments to make this a more comprehensive work. We use inductive link prediction datasets proposed in [1], including four versions of subsets based on FB15k-237 (FB237). We test NodePiece with the completely random quantization (RQ) proposed in our paper, and show the  preliminary results (Hits@10) in the table below. We will complete these results during paper revision due to the limited time.
>
> The results show that NodePiece with random entity quantization performs as well as the original model in inductive link prediction, which supports and strengthens our claim that random entity quantization is effective.
>
> | Method       | FB237-v1 | FB237-v2 | FB237-v3 | FB237-v4 |
> | ------------ | -------- | -------- | -------- | -------- |
> | NodePiece    | 0.873    | 0.939    | 0.944    | 0.949    |
> | NodePiece+RQ | 0.867    | 0.942    | 0.945    | 0.944    |
>
>
>
> Q2: In line 178, the authors mentioned that one entity matches all its connected relations. However, NodePiece randomly samples a set of $ m$ unique outgoing relations for an entity, where $m$ is a hyper parameter. It seems that the statement in this paper is in conflict with that in NodePiece.
>
> A2: We would like to clarify that what we do is exactly the same as NodePiece. Although NodePiece's paper says, "randomly sample $m$ unique outgoing relations," its actual practice in the code is to add inverse edges to KG and then select $min(m, d)$ unique relations for one entity $e$ from all its connected relations, where $m$ is a hyperparameter and $d$ is the number of all $e$'s connected relations. Our experiments and analysis follow the same setting. We will modify our statement in line 178 to make it more rigorous. This statement should be like, "After adding reverse edges, one entity matches $min(m,d)$ unique relations from its connected relations."
>
>
>
> [1] Inductive Relation Prediction by Subgraph Reasoning.

---

### Meta-Review · Area_Chair_qjQy · 2023-09-20

**Recommendation:** 5

**Metareview:**

This paper introduces the technique of random entity quantization to knowledge base embeddings. The paper is also appealing as it questions the validity of existing methods. The experimental results, inclusive of additional experiments in the discussion period, are persuasive. We eagerly await the camera-ready version that incorporates feedback from the reviews and discussions!

---

### Decision · Program_Chairs · 2023-10-07

**Decision:**

Accept-Main

**Comment:**

This paper introduces the technique of random entity quantization to knowledge base embeddings. The paper is also appealing as it questions the validity of existing methods. The experimental results, inclusive of additional experiments in the discussion period, are persuasive. We eagerly await the camera-ready version that incorporates feedback from the reviews and discussions!